# A Rare Case of Invasive Thyroid Aspergillosis Revealed on ^18^F-FDG-PET/CT

**DOI:** 10.3390/diagnostics14131451

**Published:** 2024-07-08

**Authors:** Ayoub Jaafari, Sohaïb Mansour, Laetitia Lebrun, Keitiane Kaefer, Rachid Attou

**Affiliations:** 1Nuclear Medicine Department, H.U.B Hospital, 1070 Brussel, Belgium; 2Internal Medicine Department, C.H.U Brugmann, 1020 Brussel, Belgium; sohaib.mansour@chu-brugmann.be; 3Anatomopathological Department, H.U.B Hospital, 1070 Brussel, Belgium; laetitia.lebrun@hubruxelles.be; 4Intensive Care Unit Department, C.H.U Brugmann, 1020 Brussel, Belgium; keitymed@hotmail.com (K.K.); rachid.attou@chu-brugmann.be (R.A.)

**Keywords:** invasive aspergillosis, thyroid aspergillosis, hypermetabolic thyroid nodule, ^18^F-FDG-PET/CT

## Abstract

Invasive aspergillosis (IA) represents a common form of fungal infection caused by various species of *Aspergillus* that most frequently affect immunocompromised patients. Typically, this disease occurs preferentially in high-risk groups including patients infected with the human immunodeficiency virus (HIV), patients with leukemia, patients with autoimmune diseases, and organ transplant patients undergoing medical immunosuppression. Considered the second most common cause of opportunistic fungal infection in humans after *Candida albicans*, this pathogen predominantly affects the lungs, but it may also spread by a hematogenous route to various organs and have a heterogeneous presentation. Owing to its high iodine levels, high perfusion, and enclosed capsule, the thyroid gland is considered to have a lower susceptibility to microbial invasion, and it is fairly uncommon to find associated infectious nodules. In metabolic imaging, ^18^F-FDG-PET/CT has become increasingly useful for detecting a wide range of infectious and inflammatory diseases and is already the gold standard for certain indications. According to the literature, no studies of hypermetabolic nodular thyroid aspergillosis on ^18^F-FDG-PET/CT confirmed on histology have yet been reported. Here, we report the first case of a patient with a heterogeneous presentation of IA and the presence of a hypermetabolic nodule in the thyroid with a surprising result.

**Figure 1 diagnostics-14-01451-f001:**
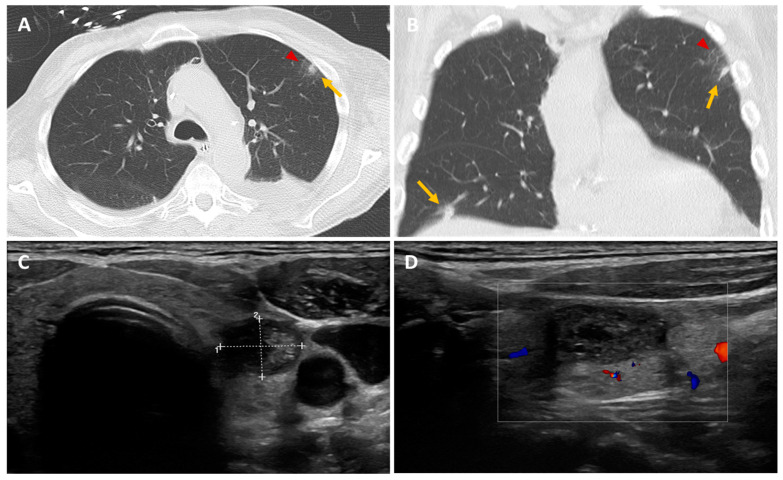
We report the case of a 71-year-old patient admitted to intensive care with severe abdominal-onset sepsis treated with broad-spectrum antibiotics. The patient had undergone liver transplantation for a moderately differentiated hepatocarcinoma operated on more than 1 year previously. During the workup, and given his immunosuppression, an extensive microbiological mapping was carried out (viral, bacterial, parasitic, fungal) which came back positive for Clostridium difficile. Antibiotic therapy was rapidly modulated in the light of our results, with a favorable clinical and biological outcome. In his follow-up, we noted the presence of a positive aspergillary galactomannan antigen, which was checked twice at 7-day intervals. A thoracic CT scan was performed and showed the presence of two pulmonary nodules (yellow arrow) in the left upper lobe with ground glass (red arrow) and the right lower lobe (**A**,**B**). A bronchoalveolar lavage (BAL) was achieved and cultured, confirming the presence of Aspergillus Fulmigatus and the diagnosis of pulmonary aspergillosis. An ^18^F-FDG-PET/CT scan was ordered to assess the presence of any extrapulmonary sites and revealed subcutaneous/muscular nodules and focal hypermetabolism in the left thyroid lobe (Figure 2, yellow arrow). In light of the results, a thyroid workup was carried out; the biological tests came back normal, and the thyroid ultrasound revealed an iso- and hypoechoic EU-TIRADS-5 macro-nodule of the left lobe measuring 18 × 12 × 23 mm, with vascularization around the perimeter, which had been punctured (**C**,**D**). Histological results showed the presence of filaments/hyphae scattered throughout in free form or septate and ramified, characteristic of aspergillosis (Figure 3, black arrow). Aspiration and culture revealed the presence of Aspergillus fulmigatus. The diagnosis of invasive thyroid aspergillosis was made. “Dot lines” and number “+” are the ultrasound size measurement tools used to measure the thyroid nodule. The “white square frame” with the colors inside refers to the vascularization of the thyroid nodule. Aspergillosis is a prevalent fungal infection caused by inhaling spores (named conidia) of the mold Aspergillus [1]. These are generally contained in soils, plants, and decaying organic matter, but also in the air and indoor environments. Over 200 species of Aspergillus are currently thought to have been described, of which only around 10% are pathogenic to humans [2]. The genus Aspergillus comprises several hundred species, of which Aspergillus fumigatus remains the most common pathogenic agent, responsible for approximately 90% of cases of aspergillosis, followed by Aspergillus flavus, niger, terreus, and nidulans [3]. In general, aspergillosis is fairly widespread. Depending on the patient’s immunity, they can lead to various forms of clinical presentation, ranging from an isolated form to multi-organ involvement [4]. The most severe forms, known as “invasive”, are usually seen in immunocompromised patients due to HIV infection, oncological conditions, intensive chemotherapy, immunosuppressive drug therapy, post-transplant conditions, or other chronic diseases, and may be associated with high mortality rates and life-threatening illness [5]. It is therefore crucial to assess both the extent of the disease and the effectiveness of treatment, with several diagnostic strategies currently available. The cornerstone of IA diagnosis is biopsy-based fungal culture, supplemented by other biomarker tests such as galactomannan in serum and bronchoalveolar lavage fluid (BAL) [6]. Conventional CT imaging is generally useful in diagnostic management, but radiological features of aspergillosis are not specific, and the CT finding termed the “halo-signal” is a sign that may not always be clearly detectable [7].

**Figure 2 diagnostics-14-01451-f002:**
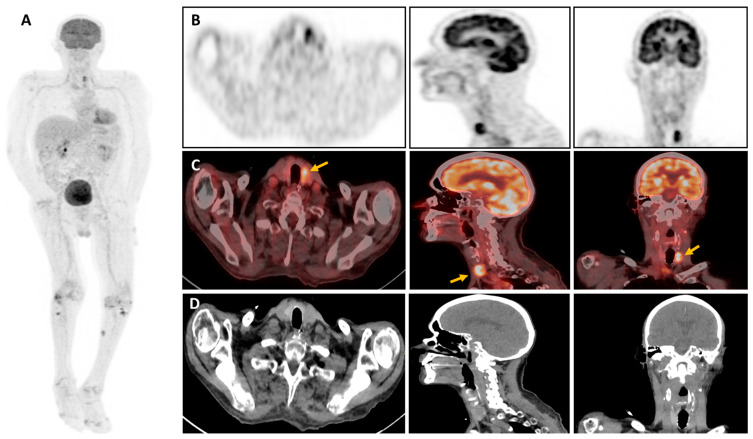
In this challenging context, nuclear medicine imaging has gained in popularity, particularly through the performance of 2-[^18^F]-fluoro-2-deoxy-D-glucose positron emission tomography combined with computed tomography (^18^F-FDG-PET/CT) in the diagnosis and monitoring of infectious diseases [8]. ^18^F-labeled FDG is a glucose analog absorbed by cells via a membrane transporter (GLUT-1 and GLUT-3) and sequestered after phosphorylation. Therefore, its intracellular concentration is proportional to the cell’s metabolism [8]. Typically, cellular use of glucose rises in leukocytes activated by infection and inflammation, leading to intense uptake of ^18^F-FDG, detectable by PET/CT [9]. ^18^F-FDG-PET/CT of the patient: (**A**) Maximum Intensity Projection (MIP) of the entire body. (**B**) Maximum 32 Intensity Projection (MIP) of the head section passes at the level of the thyroid compartment. (**C**) fused PET/CT images, 33 (**D**) CT-images showing the hypermetabolic thyroid nodule (yellow arrow). Due to the heterogeneous presentation of IA, ^18^F-FDG-PET/CT is a consistent tool, particularly in the early stages of the disease, and for the detection of extrapulmonary lesions [10]. According to the literature, the metabolic activity of aspergill nodules is not necessarily high and may vary (low to high uptake) depending on aggressiveness and form (acute or chronic IA) [11]. Some authors have reported different metabolic imaging presentations of aspergillosis lesions; a cold nodule with discrete surrounding metabolism, an iso-metabolic one when glucose uptake is similar to or less than the mediastinal blood pool, and a hypermetabolic one when glucose uptake is greater than the mediastinal blood pool as in our patient (Figure 2A,B) [11,12]. Hence, IA can result in a wide range of clinical manifestations of aspergillosis and a broad spectrum of possible presentations on ^18^F-FDG-PET/CT imaging. In our view, the thyroid gland is an unusual localization as it is extremely resistant to infection owing to its high iodine level, hydrogen peroxide production, extensive lymphatic and vascular supply, and encapsulated location [12]. Aspergillosis predominantly affects the lungs, invading the lung parenchyma and vasculature, but may also spread by a hematogenous route to various organs such as the central nervous system, heart, kidney, skin, soft tissues, and liver [5]. Once the thyroid gland is invaded, it may show variability in thyroid bioassays, ranging from hyperthyroidism due to hormone release from follicular cell lesions to hypothyroidism or euthyroidism, as in our patient [10]. According to the literature, innate and adaptive immunity have long been recognized as playing a crucial role in combating aspergillosis infection [13,14,15]. The innate immunity mediated by macrophages, neutrophils, natural killers, and plasmacytoid dendritic cells is essential for phagocytosis and fungal damage. Therefore, in our patient, it appears highly likely that the iatrogenic immunosuppression induced by the drugs used in the liver transplant significantly disrupted the patient’s immune homeostasis and led to the development of this opportunist infection, which spread from his lungs through his bloodstream to his thyroid gland. Moreover, Aspergillus fumigatus has a greater proclivity than other Aspergillus species to invade the thyroid gland and potentially induce significant destruction of thyroid tissue [16].

**Figure 3 diagnostics-14-01451-f003:**
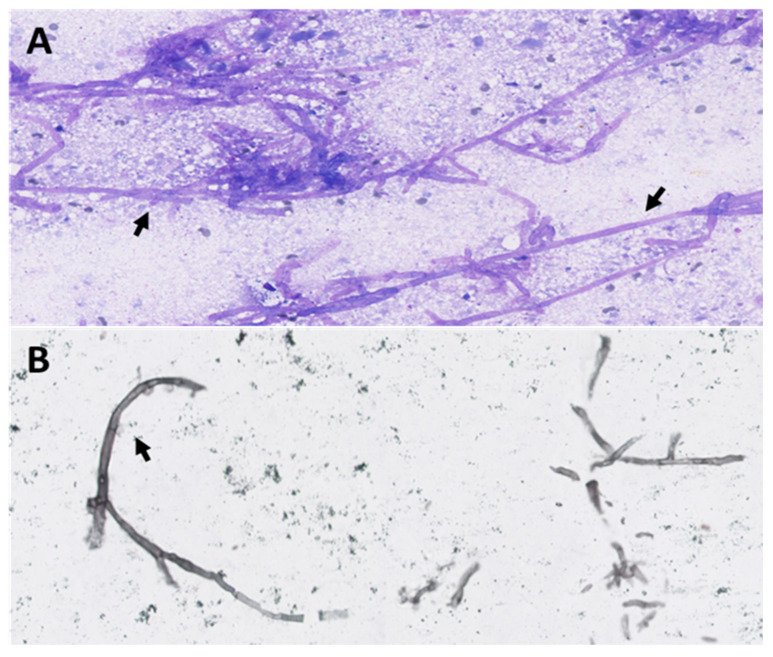
Hematoxylin–eosin (HE) (**A**) and Grocott’s (**B**) staining show the presence of filaments/hyphae scattered throughout in free form or septate and ramified, characteristic of aspergillosis (black arrow). The majority of cases are frequently asymptomatic, and given the limited epidemiological data on thyroid aspergillosis in the literature, the diagnosis is often made at autopsy (approximately 10–12% of extrapulmonary disease) [16,17]. According to the literature, no studies of hypermetabolic nodular thyroid aspergillosis on ^18^F-FDG-PET/CT confirmed on histology have yet been reported. In our patient, given the focal thyroid hypermetabolism, ultrasound and fine needle aspiration (FNA) were performed to avoid missing another pathology, revealing the presence of a hypoechoic, non-vascularized nodule (Figure 1C,D) and aspergillosis filaments (Figure 3), confirming the diagnosis of the rare thyroid invasive aspergillosis. In conclusion, invasive aspergillosis is a relatively common fungal infection in immunocompromised patients, generally affecting the lungs, but can also spread by hematogenous route and invade other organs. Thyroid aspergillosis has been rarely reported in the literature and is an entity that physicians should be aware of. The scope and severity of clinical manifestations may vary according to the individual’s immune status and the location of the invaded organs. Delays in diagnosis therefore often contribute to fatal outcomes and delayed treatment. Thyroid fine needle aspiration (FNA) is minimally invasive and has a high diagnostic efficiency. Clinicians must incorporate ^18^F-FDG-PET/CT into the diagnostic and therapeutic management of patients with suspected aspergillosis to provide a comprehensive assessment and determine the most suitable therapy.

## Data Availability

The data used and analyzed in this study are available from the corresponding author on reasonable request.

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
