# Peer review of "A Rare Case of Invasive Thyroid Aspergillosis Revealed on 18F-FDG-PET/CT"

_diagnostics, 2024, doi:10.3390/diagnostics14131451_

Round 1

Reviewer 1 Report

Comments and Suggestions for Authors

This case is rare and well-presented. I just have two minor suggestions.

1. Please change the order of figure 1 and 2. As the patient had CT first than had PET Scan.

2. The name of microbiology should be written in Italic form.

Author Response

Dear reviewer,

thank you for your feedback and your interest in our article.

The changes have been made: the images have been adjusted and the names of microbiology have been written in Italic form.

Kind regards 

Dr. JAAFARI Ayoub 

Reviewer 2 Report

Comments and Suggestions for Authors

Jaafari and colleagues from Belgium submit am “interesting image” category manuscript regarding invasive thyroid aspergillosis on 18FDG-PET/CT.

Major Comments:

-          Was the thyroid aspirate cultured? Or was it only pathology that confirmed the presence of a filamentous fungus, and you are further assuming it is based on the fact that the BAL cultured out Aspergillus fumigatus?

-          Figure 1 has odd horizontal stretching, which causes distortion. The 3 panels in the second column of the figure, there the thyroid is obvious, look as though they have been compressed horizontally. The 3 panels of the side view of the brain in the third column look as though they have been stretched horizontally. I would work on adjusting the width of each panel in Figure 1 so that each of the 10 panels is not distorted.

-          This whole case report is two long paragraphs. Information is scattered. Suggest reformatting the case report into multiple sections, and rewriting for clarity once the structure of the case report has been revamped.

-          I would not start the case report with “Figure 1”. I would start the manuscript with “We report the case of a 71-year-old patient…”. Figure 1 would appear in the text when necessary. The figures should be presented in sequential order. Right now the first figure referred to in the text is Figure 2A, 2B. So, the figures will need to be renumbered.

-          Doing a quick search on Aspergillus and the thyroid gland found more than 80 references for me, so the authors could provide an extra paragraph explaining more about the mechanism of how Aspergillus comes to settle in the thyroid gland.

-          The title “Invasive thyroid aspergillosis on 18FDG-PET/CT: When filaments are lost along the way” is meaningless. I would change it to “First case demonstrating invasive thyroid aspergillosis on 18FDG-PET/CT”.

Minor Comments:

-          Line 36: consider replacing the word “bacteriological” with “microbiological”, since more organisms than just bacteria are being sought out.

-          In general, Latin genus and species names should be italics. This would apply to Clostridium difficile, Candida albicans,

-          Line 38:  is “aspergillary antigen” actually ”Aspergillus galactomannan antigen”?

-          Line 39:  “checked 2 times”. How much time was between these two checks? A week? A month?

-          Line 40:  is “upper left lobar” actually “left upper lobe”?

-          Line 40: is “frosted glass” actually “ground glass”?

-          Line 40: is “lower right lobar” actually “right lower lobe”?

-          Line 46: is “an aspergill filament” actually “a hyphal segment from a filamentous mold”?

-          Line 48: this sentence refers to how humans develop aspergillosis, supported by reference 1. However, reference 1 is about “fungal infections”, not “aspergillus infections”, and it is dated since it was published in 2012. Suggest updating to a more appropriate reference, such as PMID 34644473.

-          Line 52: is “Aspergillus flavus, Niger, Terreus and Nidulans” actually “Aspergillus flavus, niger, terreus and nidulans”?

-          Line 82: is “shallow epidemiological data” actually “limited epidemiological data”?

-          DOIs should be added to the references.

-          Reference numbers in the text do not need to be in bold.

-          There is a problem with the reference manager. There is no reference 4 or reference 15 in the text.

Comments on the Quality of English Language

-          Line 36: consider replacing the word “bacteriological” with “microbiological”, since more organisms than just bacteria are being sought out.

-          Line 38:  is “aspergillary antigen” actually ”Aspergillus galactomannan antigen”?

-          Line 40:  is “upper left lobar” actually “left upper lobe”?

-          Line 40: is “frosted glass” actually “ground glass”?

-          Line 40: is “lower right lobar” actually “right lower lobe”?

-          Line 46: is “an aspergill filament” actually “a hyphal segment from a filamentous mold”?

-          Line 82: is “shallow epidemiological data” actually “limited epidemiological data”?

Author Response

Dear Reviewer,

Thank you for your feedback.

All major and minor corrections have been made and your comments taken into account in our rare case of thyroid aspergillosis. 
We are sending back a new version of our case, which I hope you will find satisfactory. 

Major comments: 

  • Thyroid aspiration was performed and the culture revealed the presence of Aspergillus Fulmigatus, as did the results of the BAL culture. 
  • Images were modified 
  • Paragraphs were reajusted 
  • Headline was changed 

Minor comments:

- everything was done. 

Kind regards 

Dr. JAAFARI Ayoub 

Round 2

Reviewer 2 Report

Comments and Suggestions for Authors

This revised manuscript is much improved. Most of my comments on this revised relate to language.

The species is "fumigatus". It should always be written without a capital letter and it should always be in italics.

The genus is "Aspergillus". It should always be written with a capital letter and it should always be in italics.

The infections is "aspergillosis". It is only written with a capital letter at the beginning of a sentence. It is never in italics.

The reviewer believes that "aspergillary" is not a word.

Comments on the Quality of English Language

Improved, still some changes need to be made as detailed in comments to authors.

Author Response

Dear reviewer,

Minors modifications have been made according to your commentaries

Warm Regards,

Dr JAAFARI Ayoub